# Sparse Causal Discovery with Generative Intervention for Unsupervised Graph Domain Adaptation

**Junyu Luo** [1 2]  **Yuhao Tang** [2]  **Yiwei Fu** [2]  **Xiao Luo** [3]  **Zhizhuo Kou** [4]  **Zhiping Xiao** [5]
**Wei Ju** [1 2]  **Wentao Zhang** [2]  **Ming Zhang** [1 2]

## Abstract

Unsupervised Graph Domain Adaptation (UGDA) leverages labeled source domain graphs to achieve effective performance in unlabeled target domains despite distribution shifts. However, existing methods often yield suboptimal results due to the entanglement of causal-spurious features and the failure of global alignment strategies. We propose SLOGAN (Sparse Causal Discovery with Generative Intervention), a novel approach that achieves stable graph representation transfer through sparse causal modeling and dynamic intervention mechanisms. Specifically, SLOGAN first constructs a sparse causal graph structure, leveraging mutual information bottleneck constraints to disentangle sparse, stable causal features while compressing domain-dependent spurious correlations through variational inference. To address residual spurious correlations, we innovatively design a generative intervention mechanism that breaks local spurious couplings through cross-domain feature recombination while maintaining causal feature semantic consistency via covariance constraints. Furthermore, to mitigate error accumulation in target domain pseudo-labels, we introduce a category-adaptive dynamic calibration strategy, ensuring stable discriminative learning. Extensive experiments on multiple real-world datasets demonstrate that SLOGAN significantly outperforms existing baselines.

[1]State Key Laboratory for Multimedia Information Processing, School of Computer Science, PKU-Anker LLM Lab [2]Peking University [3]University of California, Los Angeles [4]Hong Kong University of Science and Technology [5]Paul G. Allen School of Computer Science and Engineering, University of Washington. Correspondence to: Xiao Luo <xiaoluo@cs.ucla.edu>, Zhiping Xiao <patxiao@uw.edu>, Ming Zhang <mzhang_cs@pku.edu.cn>.

*Proceedings of the $42^{nd}$ International Conference on Machine Learning*, Vancouver, Canada. PMLR 267, 2025. Copyright 2025 by the author(s).

## 1. Introduction

Graph Neural Networks (GNNs) have demonstrated their effectiveness in graph-structured data analysis, including molecular structures (Turutov & Radinsky, 2024; Kim et al., 2023), social networks (Kumar et al., 2020; Li et al., 2021), traffic networks (Chowdhury et al., 2024) and RAG (Jiang et al., 2024; Luo et al., 2025). Graph classification is an important task in graph learning, which targets at predicting the labels of the entire graphs (Ying et al., 2018; Ranjan et al., 2020; Zhang et al., 2018). GNN-based graph classification approaches typically use a message-passing network in conjunction with a readout operator to aggregate the node features into graph-level representations, followed by a classifier for downstream tasks such as molecular property prediction in drug and material science (Xu et al., 2021).

Despite the effectiveness of existing approaches, their assumption of consistent distribution between training and inference datasets rarely holds, especially in real-world scenarios. This mismatch leads to out-of-distribution (OOD) challenges (Wu et al., 2020; Lin et al., 2023; You et al., 2022). In addition, the scarcity of labeled data in the target domain brings extra challenges, making supervised methods infeasible (Hao et al., 2020; Suresh et al., 2021). One typical method to tackle this problem is Unsupervised Domain Adaptation (UDA), which leverages labeled data from a source domain to enable task performance in an unlabeled target domain (Deng et al., 2021; Long et al., 2018).

UDA has been extensively studied in the context of Euclidean data (*e.g.*, images) through self-learning (Wei et al., 2021a; Xiao & Zhang, 2021), adversarial domain transfer (Ganin et al., 2016; Saito et al., 2018), or domain discrepancy minimization techniques (Lee et al., 2019a). However, it is non-trivial to develop a UDA approach for non-Euclidean graph data. There are significant challenges due to the complex topology and rich semantics of graphs (Velickovic et al., 2019; Huang & Zitnik, 2020), which are mainly attributed to: ***(1) Feature entanglement and persistent spurious correlations.*** Graph data inherently encodes both causal relationships and statistical correlations. Traditional methods that rely solely on semantic labels for model training often fail to distinguish between these two

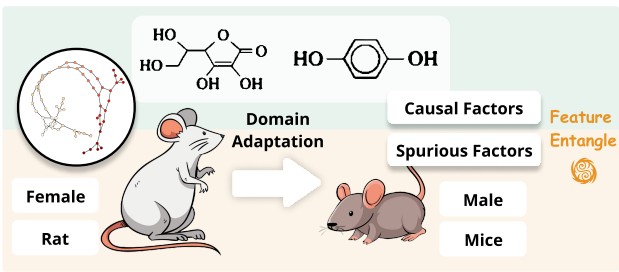

*Figure 1.* Illustration of causal and spurious factors using the PTC dataset. *Causal factors* (above) represent inherent molecular structures that directly determine carcinogenicity properties. *Spurious factors* (below) exhibit statistical correlations but lack causal relationships, potentially hindering cross-domain generalization.

types of relationships, leading to the entanglement of causal and spurious factors. For instance, in the Predictive Toxicology Challenge (PTC (Helma et al., 2001)) illustrated in Figure 1, while specific molecular structures serve as causal factors for carcinogenicity determination, experimental variables such as gender and species merely exhibit statistical correlations with the labels. Without explicit disentanglement, these spurious factors can interfere with cross-domain generalization through residual correlations, causing the model to rely on unstable mechanisms in the target domain. *(2) Alignment collapse and global strategy failure.* Existing methods typically rely on adversarial learning to align domain distributions (Dai et al., 2022; Shen et al., 2020). However, their global alignment strategy can lead to information collapse, discarding crucial carriers like rare substructures, while failing to suppress spurious factors effectively. These challenges are particularly severe in graph data due to their complex structure and high-dimensional sparsity, necessitating explicit intervention mechanisms.

To address the above challenges in graph domain adaptation, we propose **S**parse Causal Disc**o**very with **G**enerative Interventio**n** (SLOGAN) for Unsupervised Graph Domain Adaptation. SLOGAN leverages sparse causal discovery and dynamic intervention learning to extract stable causal mechanisms while suppressing spurious correlations. Our framework consists of three key components: *First,* SLOGAN focuses on causal-spurious feature disentanglement, based on the structural causal model and Information Bottleneck (IB) principle, we decompose graph representations into causal and spurious features. This decomposition compresses label-irrelevant spurious information while preserving sparse, stable causal patterns. *Second,* SLOGAN introduces a generative intervention mechanism, to avoid global alignment collapse. We design a generative model to reconstruct original graph representations with a cross-domain spurious feature exchange strategy. By perturbing local coupling of spurious features, this approach forces the model to rely solely on causal features for reconstruction, effectively suppressing spurious residuals. *Third,* SLOGAN introduces

dynamic stability optimization to address error propagation risks in target domain pseudo-labels. We propose category-adaptive confidence thresholds for dynamic and balanced sample selection. Through cross-domain joint optimization, SLOGAN achieves progressive alignment of causal features, enhancing the framework's robustness. Extensive experiments on 6 benchmark datasets demonstrate SLOGAN's consistently superior performance over existing methods.

The contribution of this paper is summarized as follows. **(1)** We focus on sparse stability and dynamic robustness in UGDA, proposing a framework based on stable learning and causal intervention that effectively extracts domain-invariant representations. **(2)** We introduce SLOGAN, which introduces stable causal feature extraction with generative interventions and adaptive pseudo-label calibration to achieve high-performance. **(3)** We provide theoretical guarantees for the optimization error bound in the target domain. **(4)** We conduct extensive experiments to demonstrate SLOGAN's superior performance and varfied our motivation.

## 2. Preliminaries

***Unsupervised Graph Domain Adaptation.*** For each graph sample $\mathcal{G} = (\mathcal{V}, \mathcal{E})$, $\mathcal{V}$ denotes the set of nodes and $\mathcal{E} \subseteq \mathcal{V} \times \mathcal{V}$ represents the edge set. The node feature matrix is $X \in \mathbb{R}^{|\mathcal{V}| \times d}$, where each row $x_v \in \mathbb{R}^d$ is the feature representation for node $v \in \mathcal{V}$. $d$ denotes the dimension of node features. A labeled source dataset is $\mathcal{D}_{so} = \{(\mathcal{G}_i^{so}, y_i^{so})\}_{i=1}^{N_{so}}$, comprising $N_{so}$ graph examples $\mathcal{G}_i^{so}$ with corresponding labels $y_i^{so}$. An unlabeled target dataset is $\mathcal{D}_{ta} = \{\mathcal{G}_j^{ta}\}_{j=1}^{N_{ta}}$, which contains $N_{ta}$ graph examples $\mathcal{G}_j^{ta}$ without label. These two domains have the same label space $\{1, \cdots, C\}$; however, their data distributions have a huge gap. Our goal is to transfer knowledge from the labeled source graphs to the unlabeled target graphs.

***Graph Neural Networks.*** Given a graph $\mathcal{G} = (\mathcal{V}, \mathcal{E})$ with node features $X$, a GNN learns node representations via message passing. For each node $v$ at layer $l$:

$$h_v^{(l)} = \text{UPDATE}\left(h_v^{(l-1)}, \text{AGG}\left(\{h_u^{(l-1)} \mid u \in \mathcal{N}(v)\}\right)\right), \tag{1}$$

where AGG aggregates neighbor features and UPDATE updates node features. The final graph representation is:

$$\mathbf{p} = \text{CLA}\left(\text{READOUT}\left(\{h_v^{(L)} \mid v \in \mathcal{V}\}\right)\right), \tag{2}$$

where READOUT pools node features and CLA generates predictions. The model is trained with cross-entropy loss:

$$\mathcal{L}_{so} = -\frac{1}{|\mathcal{D}^{so}|} \sum_{\mathcal{G}_i^{so} \in \mathcal{D}^{so}} \log \mathbf{p}_i^{so}[y_i^{so}]. \tag{3}$$

***Motivation & Challenges.*** Existing UGDA methods face

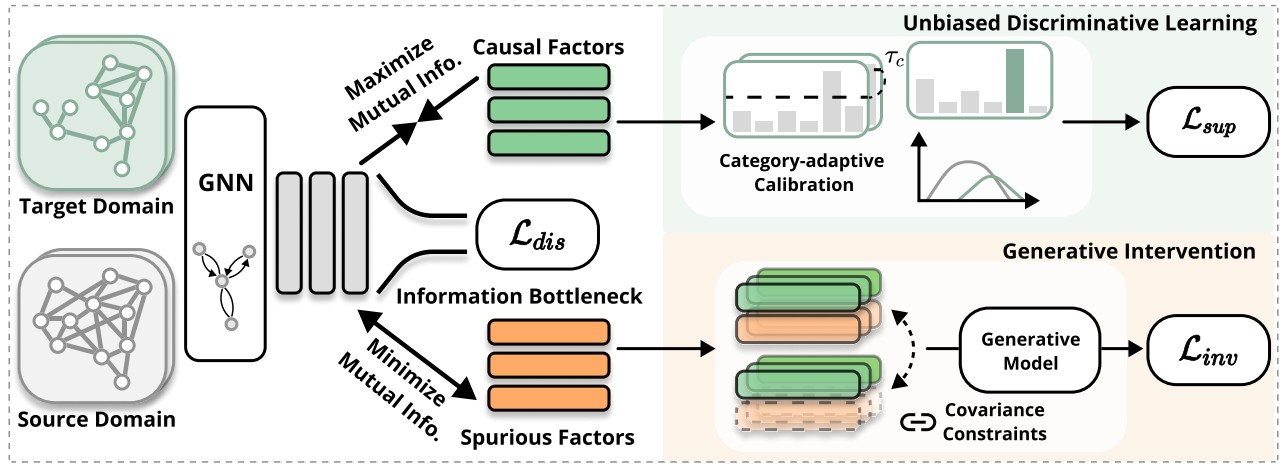

*Figure 2.* Overview of SLOGAN. The framework consists of three key components: (1) A structural causal model to disentangle causal and spurious factors from graph representations. (2) An unbiased discriminative learning module with category-adaptive calibration. (3) A generative intervention mechanism with covariance constraints to suppress the influence of spurious factors across domains.

two fundamental limitations. First, deep entanglement of causal and spurious features leads models to rely on domain-specific correlations, compromising cross-domain generalization. Second, global alignment strategies often cause alignment collapse by disrupting critical causal substructures while preserving spurious couplings. These challenges necessitate explicit causal-spurious disentanglement and local stability preservation, which motivates our sparse causal graph construction and generative intervention mechanism.

## 3. Methodology

### 3.1. Framework Overview

In this paper, we introduce SLOGAN, as shown in Fig. 2, which achieves reliable knowledge transfer through three complementary stability-enhancing components: (1) Sparse causal discovery through feature disentanglement, (2) Progressive stable alignment via discriminative learning with confidence calibration, and (3) Generative intervention with covariance constraints for spurious feature suppression.

### 3.2. Sparse Causal Discovery via Feature Disentanglement

We address domain shift in UGDA through sparse stability learning, where the key challenge lies in identifying stable causal features ($Z^c$) that remain predictive across domains while suppressing unstable spurious features ($Z^s$). Our approach implements sparse variable independence (SVI) through three stability-enforcing mechanisms:

***Stable Causal Graph Construction.*** As illustrated in Fig. 3, the structural causal model establishes sparse dependencies between variables to ensure stability (Yu et al., 2023). The graph depicts key causal relationships where $L$ represents

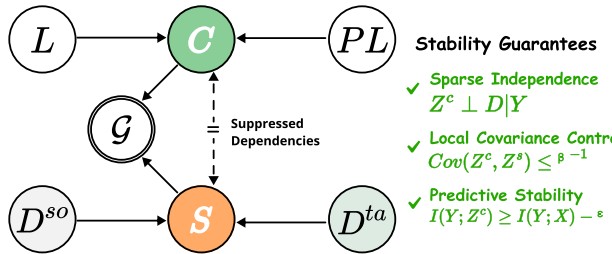

*Figure 3.* Causal graph for UGDA. Dashed arrows indicate suppressed dependencies through sparse stability constraints.

labels, $C$ denotes causal features, $S$ is spurious features, and $PL$ represents pseudo-labels. The dashed arrows highlight suppressed dependencies through stability constraints. This causal structure enables three key mechanisms:

- Sparse Feature Generation: $C \rightarrow \mathcal{G} \leftarrow S$ enforces minimal sufficient causal pathways while allowing residual spurious correlations

- Label Stability: $L \rightarrow C \leftarrow PL$ creates cross-domain stability through pseudo-label constrained learning

- Domain Sparsity: $D^{so} \rightarrow S \leftarrow D^{ta}$ isolates domain-specific variations to spurious features

***Stability-Aware Disentanglement Learning.*** Using GNN-derived features $\mathbf{z}$, we implement sparse variable independence (SVI) through:

$$\underbrace{\max I(Y; Z^c)}_{\text{Stable Prediction}} - \underbrace{\beta I(Z^s; Z)}_{\text{Residual Control}} + \underbrace{\min I(Z^s; Y)}_{\text{Spurious Suppression}} . \quad (4)$$

This objective comprises three key components: (1) maximizing mutual information between causal features and labels ensures stable prediction across domains, (2) control-

ling residual information between spurious features and original representations prevents feature collapse, and (3) minimizing mutual information between spurious features and labels reduces the impact of domain-specific variations.

***Causal Feature Extraction.*** To improve the extraction of causal features, the objective is to maximize the mutual information between the causal feature $\mathbf{z}^c$ and the semantic label $\mathbf{y}$ (Wang et al., 2024a). We employs InfoNCE (He et al., 2020), which samples positive pairs from the joint distribution $p(\mathbf{z}^c, \mathbf{y})$ and negative pairs from the marginal distributions $p(\mathbf{z}^c) p(\mathbf{y})$. We disentangle the causal feature $\mathbf{z}^c$ with:

$$\min \mathcal{L}_{MI}^c = \mathbb{E}_{p(\mathbf{z}^c, \mathbf{y})}[\xi] - \log \mathbb{E}_{p(\mathbf{z}^c)p(\mathbf{y})}[e^\xi], \quad (5)$$

where $\xi = F^c(\mathbf{z}^c, \mathbf{y})$ implements a bilinear mapping $F^c(\mathbf{z}^c, \mathbf{y}) = \mathbf{z}^{cT} W \mathbf{y}$ with learnable parameter matrix $W$. This bilinear form captures pairwise interactions between causal features and labels while maintaining computational efficiency.

***Spurious Feature Suppression.*** To effectively disentangle spurious features, we minimize the mutual information between spurious features $\mathbf{z}^s$ and semantic labels $\mathbf{y}$ while maintaining residual information through a constrained optimization:

$$\min \mathcal{L}_{MI}^s = I(\mathbf{z}^s, \mathbf{y}) - \beta I(\mathbf{z}^s, \mathbf{z}), \quad (6)$$

where $\beta$ balances the competing objectives. We adopt the Variational Information Bottleneck (VIB) framework due to its effectiveness in learning compressed representations while maintaining relevant information. Using VIB, we derive variational bounds to control spurious correlations:

$$I(\mathbf{z}^s, \mathbf{y}) \le \mathbb{E}_{p(\mathbf{z}^s, \mathbf{y})}[\log q(\mathbf{y}|\mathbf{z}^s)] - \mathbb{E}_{p(\mathbf{z}^s)p(\mathbf{y})}[\log q(\mathbf{y}|\mathbf{z}^s)],$$
$$(7)$$

where $q$ approximates the true conditional distribution. For the mutual information between spurious and graph-level features, we employ a bilinear estimator $\psi$ to derive:

$$I(\mathbf{z}^s, \mathbf{z}) \le \mathbb{E}_{p(\mathbf{z}^s, \mathbf{z})}[\psi] - \log(\mathbb{E}_{p(\mathbf{z}^s)p(\mathbf{z})}[e^\psi]). \quad (8)$$

The reconstruction constraint $I(\mathbf{z}^s, \mathbf{z}) \le I_c$ prevents complete feature collapse while breaking domain-specific couplings, enabling effective spurious feature suppression during adaptation.

***Stability Guarantees.*** Our formulation ensures three fundamental stability properties. **First, sparse independence** ($Z^c \perp D|Y$) is achieved through domain-invariant contrastive learning, ensuring that causal features remain independent of domain shifts conditioned on labels. **Second, local covariance control** maintains $\mathrm{Cov}(Z^c, Z^s) \le \beta^{-1}$ via Lagrange optimization, preventing excessive entanglement

between causal and spurious features while allowing necessary correlations. **Third, predictive stability** is guaranteed as $I(Y; Z^c) \ge I(Y; X) - \epsilon$ through feature compression, where $\epsilon$ controls information loss during causal feature extraction, ensuring that causal features preserve essential predictive information from the input.

The composite stability objective $\mathcal{L}_{dis} = \mathcal{L}_{MI}^c + \mathcal{L}_{MI}^s$ is minimized to disentangle causal features from spurious ones, ensuring domain-invariant representations while preserving semantic information.

### 3.3. Unbiased Discriminative Learning

To overcome target domain data scarcity while preventing overconfidence in pseudo-labels (Karim et al., 2022), we develop a stability-preserving discriminative learning mechanism. Our approach combines causal feature reliability with adaptive sample selection to maintain class-balanced supervision:

We first leverage causal features to determine the distribution of graph classification,

$$\mathbf{p}_i^{ta} = \phi\left(\mathbf{z}_i^{c\,ta}\right), \quad (9)$$

where $\phi$ projects causal features $\mathbf{z}^{c\,ta}$ to label space. We quantify prediction certainty through maximum class probability:

$$s_i^{ta} = \max_c \mathbf{p}_i^{ta}[c], \quad (10)$$

where $s_i^{ta}$ is the confidence score. Subsequently, to achieve unbiased pseudo-label estimation, we introduce adaptive confidence thresholds. The class-adaptive coefficients are:

$$\mathcal{M}_c = \max\{s_i^{ta} | \arg\max_{c'} \mathbf{p}_i^{ta}[c'] = c\}. \quad (11)$$

Therefore, the class-unbiased threshold for class $c$ is:

$$\tau_c = \mathcal{M}_c \cdot \tau, \quad (12)$$

where $\tau$ is the initial confidence score threshold, which is set to 0.95 following (Sohn et al., 2020). Then, we obtain the refined confident set $\mathcal{C}$ as:

$$\mathcal{C} = \{G_i^{ta} | c = \arg\max_{c'} \mathbf{p}_i^{ta}, s_i^{ta} > \tau_c\}. \quad (13)$$

Cross-domain stability is enforced through optimization:

$$\mathcal{L}_{ta} = -\frac{1}{|\mathcal{C}|} \sum_{\mathcal{G}_i^{ta} \in \mathcal{C}} \log \mathbf{p}_i^{ta}[\hat{y}_i^{ta}], \quad (14)$$

where $\hat{y}_i^{ta}$ denotes the pseudo-label of $\mathcal{G}_i^{ta}$. We also optimize source data using Eqn. 3 with causal features to mitigate overconfidence. The supervised loss is $\mathcal{L}_{sup} = \mathcal{L}_{so} + \mathcal{L}_{ta}$. This dual-domain strategy ensures: (1) causal feature preservation via $\mathcal{L}_{so}$, (2) error-resistant adaptation via $\mathcal{L}_{ta}$, and (3) class-balanced learning via adaptive thresholding, complementing the feature disentanglement from Section 3.2.

## 3.4. Generative Intervention with Covariance Constraints

Our method uses a targeted approach to ensure models don't rely on misleading patterns that vary across domains. Consider social network analysis: when classifying discussion topics, our method can distinguish between fundamental network structures (like community clusters and information flow patterns) and platform-specific features (like temporary trending hashtags or regional engagement patterns).

We achieve this through a generative intervention mechanism that deliberately exchanges domain-specific features between samples while preserving essential structural patterns. This forces the model to focus only on truly predictive patterns that work consistently across different environments, establishing two key stability properties: (1) independence from spurious domain-specific correlations through controlled feature swapping, and (2) preservation of essential causal patterns through reconstruction consistency.

Specifically, we use a generative model $G(\cdot, \cdot)$ to reconstruct graph representations based on causal and spurious features. The reconstruction model is a two-layer MLP. The reconstruction function is optimized by $L2$ distance as:

$$\mathcal{L}_{ge} = \mathbb{E} \left\| \mathbf{z} - G\left(\mathbf{z}^c, \mathbf{z}^s\right) \right\|_2^2 . \tag{15}$$

Here, $\mathbf{z}$ is the graph representation, $\mathbf{z}^c$ and $\mathbf{z}^s$ are the causal and spurious features. During the adaptation process, to break local correlations, we swap the spurious parts of samples from different domains within a mini-batch $\mathcal{B}$, generating new composite samples:

$$\mathbf{z}_i^{+k} = G\left(\mathbf{z}_i^c, \mathbf{z}_k^s\right) , \tag{16}$$

where $\mathbf{z}_i^{+k}$ represents the composite representation combining causal features $\mathbf{z}_i^c$ from sample $i$ with spurious features $\mathbf{z}_k^s$ from sample $k$ in different domains. Then, we enforce intervention invariance through dual constraints:

$$\mathcal{L}_{inv} = \mathcal{L}_{re} + \mathbb{E}_{\mathbf{z}_i \in \mathcal{B}^{so}, \mathbf{z}_k \in \mathcal{B}^{ta}} \left\| \mathbf{z}_i^{+k} - \mathbf{z}_i \right\|^2 , \tag{17}$$

where $\mathcal{L}_{re}$ is the reconstruction loss. Using invariant learning under intervention, we achieve robust cross-domain representations by eliminating spurious factors. The reconstruction loss $\mathcal{L}_{re}$ maintains reconstruction capability while the intervention term ensures robustness. This achieves: (1) elimination of domain-specific spurious couplings via feature recombination, and (2) preservation of causal features through consistent predictions.

***Overall Optimization.*** In a nutshell, the overall adaptation training objective is:

$$\mathcal{L} = \mathcal{L}_{sup} + \gamma \mathcal{L}_{dis} + \eta \mathcal{L}_{inv} . \tag{18}$$

where $\gamma$ and $\eta$ serve as hyperparameters to balance the contributions of the submodules, which is anlyzed in Section 4.6. Here, we first warm up our framework on source samples. Afterwards, the target domain is considered for optimization. The summarized overall algorithm can be found in the Appendix A.

## 3.5. Theoretical Guarantees

Our theoretical analysis establishes a probabilistic bound on the target domain error through three stability conditions.

**Theorem 3.1.** *Under a stable causal graph construction, assume the following conditions hold: (1)* ***Causal Sufficiency****: $I(Y; Z^c) > I_c$, where $Z^c$ is the causal variable and $I_c$ is an information contraint. (2)* ***Spurious Suppression****: $I(Y; Z^s) \leq \epsilon_1$, where $Z^s$ is the spurious variable. (3)* ***Generative Intervention****: $\mathbb{E}\|Z - G(Z^c, Z^s)\|_2^2 \leq \epsilon_2$, where $G$ is the generation model. Then, for any predictor $h \in \mathcal{H}$, with probability at least $1 - \delta$, the target domain error $\epsilon_T(h)$ is bounded as follows:*

$$\epsilon_T(h) \leq \hat{\epsilon}_S(h) + C\sqrt{\epsilon_1} + L\sqrt{\epsilon_2} + C(n_S, \delta), \tag{19}$$

*where $L$ is the lipschitz constant of the loss function, $C$ is a constant, and $n_S$ is the sample size in the source domain. Here, $\epsilon_T(h)$ represents the error in the target domain, while $\hat{\epsilon}_S(h)$ denotes the empirical error in the source domain.*

The proof sketch follows three key steps: (1) bounding feature reconstruction error via the generator's fidelity, (2) quantifying spurious correlation suppression in domain discrepancy through mutual information constraints, and (3) incorporating statistical generalization error to establish a bound through empirical risk. Detailed proof is provided in Appendix B.

***Complexity Analysis.*** The computing complexity is mainly introduced by GNN. For given graph $\mathcal{G} = (\mathcal{V}, \mathcal{E})$, $d$ is the feature dimension, $|V|$ denotes the number of nodes, and $L$ represents the number of GNN layers. The complexity of our GNN is $\mathcal{O}(L|V|d^2)$, i.e., linear to $|V|$.

# 4. Experiments

## 4.1. Experimental Settings

### 4.1.1. DATASETS

We investigate unsupervised graph domain adaptation leveraging benchmark datasets, adopting both *cross-dataset* and *dataset split* scenarios for a thorough evaluation. *For cross-dataset scenarios* we achieve domain adaptation on PTC (Helma et al., 2001). PTC are inherently unbiased across sub-datasets. *For dataset-split scenarios,* we follow previous works (Ding et al., 2018; Yin et al., 2022; Lu et al., 2023) to split the dataset by graph density. The dataset splitting experiments are performed on the TWITTER-Real-Graph-Partial (Pan et al., 2015), NCI1 (Wale & Karypis, 2006) and Letter-Med (Riesen & Bunke, 2008), using their

*Table 1.* The classification results (in %) on PTC (source→target).

| Methods | MR→MM | MM→MR | MR→FM | FM→MR | MR→FR | FR→MR | MM→FM | FM→MM | MM→FR | FR→MM | FM→FR | FR→FM | Avg. |
|---|---|---|---|---|---|---|---|---|---|---|---|---|---|
| GIN | 61.8 | 64.3 | 57.7 | 56.5 | 45.7 | 53.5 | 37.7 | 42.6 | 44.5 | 59.4 | 66.2 | 54.3 | 53.7 |
| GCN | 63.2 | 62.9 | 66.2 | 55.1 | 45.7 | 67.6 | 62.3 | 54.4 | 64.8 | 58.0 | 60.3 | 52.0 | 59.4 |
| GAT | 60.0 | 46.0 | 70.7 | 57.1 | 46.3 | 65.9 | 54.5 | 53.8 | 53.2 | 69.0 | 65.9 | 51.7 | 57.8 |
| SAGE | 58.8 | 55.1 | 67.6 | 56.5 | 47.4 | 66.2 | 48.1 | 48.2 | 45.1 | 58.0 | 63.2 | 52.6 | 55.6 |
| MeanTeacher | 61.8 | 61.4 | 73.2 | 60.9 | 52.9 | 50.7 | 65.2 | 44.1 | 35.1 | 66.7 | 55.9 | 42.9 | 55.9 |
| InfoGraph | 63.2 | 60.0 | 66.2 | 59.4 | 48.6 | 67.6 | 55.1 | 56.4 | 64.8 | 63.8 | 69.1 | 54.3 | 60.7 |
| DANN | 67.5 | 64.3 | 69.0 | 63.8 | 55.7 | 73.2 | 50.7 | 48.5 | 66.2 | 71.0 | 70.6 | 52.9 | 62.8 |
| ToAlign | 70.5 | 45.7 | 67.6 | 66.7 | 54.3 | 67.6 | 58.0 | 50.0 | 67.6 | 71.0 | 76.5 | 55.7 | 62.6 |
| DUA | 62.0 | 63.1 | 73.6 | 52.2 | 54.3 | 63.4 | 53.7 | 55.9 | 60.6 | 69.4 | 63.2 | 54.3 | 60.5 |
| DARE-GRAM | 61.8 | 54.3 | 73.3 | 53.6 | 55.7 | 66.2 | 56.4 | 54.1 | 59.4 | 69.6 | 66.2 | 55.7 | 60.5 |
| CoCo | 65.1 | 63.8 | 73.0 | 60.3 | 55.2 | 72.8 | 62.1 | 55.9 | 63.2 | 70.5 | 70.1 | 54.2 | 63.8 |
| MTDF | 65.9 | 64.8 | 76.9 | 61.2 | **56.0** | **73.9** | 62.6 | 56.8 | 69.0 | 71.1 | 71.6 | 56.0 | 65.5 |
| **Ours** | **71.1** | **65.7** | **74.6** | **66.8** | 55.4 | 71.8 | **66.7** | **59.1** | **70.4** | **78.3** | **76.6** | **57.1** | **67.8** |

*Table 2.* The classification results (in %) on NCI1 (source→target). N0, N1, N2, and N3 are sub-datasets.

| Methods | N0→N1 | N0→N2 | N0→N3 | N1→N0 | N1→N2 | N1→N3 | N2→N0 | N2→N1 | N2→N3 | N3→N0 | N3→N1 | N3→N2 | Avg. |
|---|---|---|---|---|---|---|---|---|---|---|---|---|---|
| GIN | 66.0 | 60.6 | 50.3 | 68.0 | 68.4 | 69.9 | 61.0 | 65.6 | 73.1 | 48.3 | 59.4 | 62.9 | 62.8 |
| GCN | 55.8 | 59.1 | 54.0 | 73.3 | 65.0 | 70.7 | 73.5 | 60.7 | 70.2 | 67.8 | 54.5 | 55.1 | 63.3 |
| GAT | 63.4 | 60.0 | 41.7 | 70.1 | 68.2 | 70.1 | 73.2 | 63.1 | 69.3 | 56.6 | 56.3 | 60.5 | 62.7 |
| SAGE | 54.9 | 55.8 | 50.1 | 74.5 | 59.7 | 66.0 | 76.2 | 59.7 | 71.7 | 70.6 | 57.2 | 64.8 | 63.4 |
| MeanTeacher | 54.9 | 45.2 | 51.6 | 73.8 | 45.2 | 50.7 | 73.3 | 54.9 | 50.2 | 72.8 | 55.8 | 47.1 | 56.3 |
| InfoGraph | 66.5 | 61.0 | 57.6 | 62.7 | 64.6 | 64.1 | 75.7 | 62.6 | 67.1 | 69.9 | 60.7 | 50.2 | 63.6 |
| DANN | 64.1 | 58.7 | 45.6 | 76.2 | 69.8 | 63.6 | 71.3 | 70.9 | 70.0 | 70.4 | 58.3 | 67.5 | 65.5 |
| ToAlign | 65.5 | 61.7 | 47.1 | 73.3 | 69.9 | 59.7 | 71.4 | 69.9 | 69.9 | 68.0 | 59.2 | 63.1 | 64.9 |
| DUA | 69.9 | 60.7 | 58.5 | 71.3 | 69.9 | 68.4 | 67.5 | 68.0 | 70.9 | 56.1 | 50.5 | 66.5 | 64.9 |
| DARE-GRAM | 69.4 | 59.2 | 55.8 | 69.9 | 69.4 | 61.2 | 68.9 | 70.4 | 68.9 | 60.1 | 57.6 | 65.0 | 64.7 |
| CoCo | 70.9 | 64.0 | 68.7 | 70.0 | 68.5 | 71.2 | 75.1 | 61.2 | 72.8 | 74.6 | 59.6 | 56.4 | 67.7 |
| MTDF | 67.5 | 70.9 | **71.8** | **76.7** | 65.0 | 73.1 | 77.2 | 62.5 | 74.3 | 75.9 | 61.0 | 57.8 | 69.5 |
| **Ours** | **71.4** | **64.1** | 63.1 | 71.8 | **72.3** | **72.8** | **76.7** | **72.5** | **73.3** | **76.3** | **61.7** | **70.9** | **70.6** |

diverse nature to evaluate our domain adaptation capability. The details and statistics of the datasets are provided in the Appendix C.1.

### 4.1.2. BASELINES

We employ a wide range of state-of-the-art baselines: *(1) Graph neural networks,* including GCN (Welling & Kipf, 2016), GIN (Xu et al., 2018), GAT (Veličković et al., 2018), and GraphSAGE (Hamilton et al., 2017). *(2) Semi-supervised graph methods,* including InfoGraph (Sun et al., 2020) and Mean-Teacher (Tarvainen & Valpola, 2017). *(3) Domain adaptation methods,* including ToAlign (Wei et al., 2021b), DANN (Ganin et al., 2016), DUA (Mirza et al., 2022) and DARE-GRAM (Nejjar et al., 2023). *(4) Graph adaptation methods,* including CoCo (Yin et al., 2023) and MTDF (Tang et al., 2024), the latest state-of-the-art method for UGDA. More details are available in the Appendix C.2.

### 4.1.3. IMPLEMENTATION DETAILS

The baseline methods follow the same settings as the original papers. We use a 2 layer GCN encoder with a hidden dimension of 128. We use Adam optimizer for 100 epochs source domain training with a learning rate of 0.001 and batch size of 128. For the target domain, adaptation is per-

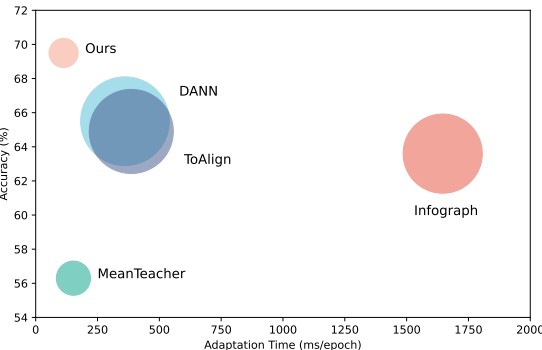

*Figure 4.* Scalability analysis on NCI1. Circle radius indicates parameter count. SLOGAN achieves best performance with minimal latency and parameters.

formed over 30 epochs. The loss weight, $\gamma$ and $\eta$, are set to 0.003 and 0.1, according to the sensitivity experiments. More details are available in the Appendix C.1.

### 4.2. Performance Comparison

In this section, we compare SLOGAN with baselines, as shown in Table 1, 3 and 2. The observations are as follows: (1) Our experiments demonstrate a notable advantage of domain adaptation methods in most cases. This highlights the

*Table 3.* The classification results (in %) on TWITTER-Real-Graph-Partial (source→target). T0, T1, T2, and T3 are sub-datasets.

| Methods | T0→T1 | T0→T2 | T0→T3 | T1→T0 | T1→T2 | T1→T3 | T2→T0 | T2→T1 | T2→T3 | T3→T0 | T3→T1 | T3→T2 | Avg. |
|---|---|---|---|---|---|---|---|---|---|---|---|---|---|
| GIN | 59.7 | 62.8 | 60.4 | 64.2 | 62.2 | 61.3 | 61.7 | 63.2 | 61.0 | 62.3 | 61.8 | 62.4 | 61.9 |
| GCN | 62.0 | 62.9 | 59.7 | 64.1 | 63.4 | 59.8 | 64.2 | 62.8 | 60.5 | 62.7 | 61.4 | 62.7 | 62.2 |
| GAT | 60.6 | 63.2 | 60.0 | 63.1 | 61.6 | 59.8 | 63.5 | 61.6 | 59.5 | 63.4 | 62.1 | 63.7 | 61.9 |
| SAGE | 61.0 | 64.6 | 62.1 | 61.9 | 61.9 | 60.8 | 62.9 | 62.6 | 60.9 | 61.7 | 60.9 | 63.4 | 62.1 |
| MeanTeacher | 52.2 | 49.2 | 46.1 | 49.0 | 50.7 | 46.1 | 49.5 | 51.7 | 52.6 | 48.1 | 48.0 | 51.1 | 49.5 |
| InfoGraph | 63.9 | 65.1 | 61.6 | 65.6 | 65.0 | 59.2 | 64.3 | 63.3 | 60.8 | 63.3 | 62.4 | 63.3 | 63.2 |
| DANN | 58.4 | 60.0 | 58.0 | 59.0 | 59.4 | 57.4 | 57.7 | 58.1 | 58.4 | 58.2 | 57.9 | 60.4 | 58.6 |
| ToAlign | 58.6 | 59.5 | 55.5 | 57.7 | 58.1 | 56.1 | 56.3 | 57.2 | 57.8 | 57.7 | 57.6 | 60.2 | 57.7 |
| DUA | 64.2 | 64.8 | 62.1 | 65.8 | 56.0 | 62.0 | 65.3 | 63.6 | 60.8 | 64.2 | 63.4 | 64.7 | 63.1 |
| DARE-GRAM | 61.7 | 65.7 | 61.6 | 65.7 | 64.4 | 62.1 | 64.0 | 64.3 | 60.0 | 64.9 | 64.2 | 64.3 | 63.6 |
| CoCo | 64.5 | 66.1 | 62.1 | 64.0 | 64.6 | 61.6 | 64.5 | 63.5 | 62.1 | 62.8 | 62.5 | 63.5 | 63.5 |
| MTDF | 64.5 | 66.4 | **65.1** | 64.7 | 65.2 | 62.2 | 64.9 | 63.5 | **63.2** | 63.2 | 63.4 | 64.4 | 64.2 |
| **Ours** | **65.1** | **66.5** | 62.3 | **66.2** | **65.4** | **62.4** | **66.6** | **64.4** | 62.2 | **65.8** | **64.4** | **65.1** | **64.7** |

*Table 4.* The classification results (in %) on Letter-Med (source→target). L0, L1, L2, and L3 are sub-datasets.

| Methods | L0→L1 | L0→L2 | L0→L3 | L1→L0 | L1→L2 | L1→L3 | L2→L0 | L2→L1 | L2→L3 | L3→L0 | L3→L1 | L3→L2 | Avg. |
|---|---|---|---|---|---|---|---|---|---|---|---|---|---|
| GIN | 54.0 | 54.0 | 47.8 | 31.0 | 48.7 | 46.0 | 23.9 | 38.1 | 58.4 | 19.5 | 26.5 | 51.3 | 41.6 |
| GCN | 57.2 | 59.3 | 59.6 | 43.4 | 51.9 | 51.9 | 40.4 | 54.3 | 51.9 | 45.1 | 34.5 | 60.8 | 50.9 |
| GAT | 79.6 | 71.7 | 67.3 | 61.9 | 75.2 | 76.1 | 55.8 | 78.8 | 75.2 | 54.0 | 69.9 | 70.8 | 69.7 |
| SAGE | 81.4 | 71.7 | 66.4 | 66.4 | 78.8 | 70.8 | 51.3 | 77.0 | 72.6 | 50.4 | 65.5 | 77.9 | 69.2 |
| MeanTeacher | 70.8 | 73.5 | 58.4 | 62.8 | 76.1 | 57.5 | 38.1 | 75.2 | 51.3 | 54.0 | 54.0 | 56.6 | 60.7 |
| InfoGraph | 80.5 | 66.4 | 62.8 | 51.3 | 74.3 | 69.0 | 42.5 | 63.7 | 62.0 | 54.3 | 59.3 | 69.0 | 62.9 |
| DANN | 70.8 | 73.3 | 53.1 | 53.1 | 73.5 | 50.4 | 38.9 | 77.9 | 60.2 | 50.4 | 61.1 | 63.7 | 60.5 |
| ToAlign | 74.3 | 72.5 | 62.8 | 64.6 | 76.1 | 50.4 | **62.8** | 73.5 | 59.3 | 50.6 | 54.9 | 54.0 | 63.0 |
| DUA | 79.6 | 71.7 | 67.3 | 61.9 | 75.2 | 76.1 | 55.8 | 78.6 | 75.2 | 54.8 | 69.9 | 70.8 | 69.7 |
| DARE-GRAM | 81.4 | 71.7 | 66.4 | 66.4 | 78.8 | 70.8 | 51.3 | 77.0 | 72.6 | 50.4 | 65.5 | 77.9 | 69.2 |
| CoCo | 76.5 | 70.3 | 68.8 | 68.3 | **80.2** | 72.8 | 55.3 | 77.6 | 75.3 | 59.5 | 68.1 | 79.2 | 71.0 |
| MTDF | 78.8 | 72.1 | 68.0 | 68.1 | 79.0 | 71.3 | 56.9 | 75.2 | 78.1 | 60.0 | 69.9 | 78.7 | 71.3 |
| **Ours** | **83.2** | **73.6** | **69.9** | 68.1 | 79.6 | **77.0** | 57.5 | **78.8** | **80.1** | 60.2 | **74.3** | **79.6** | **73.5** |

challenge of the UGDA task. (2) Semi-supervised methods, such as InfoGraph, generally outperform better. However, their relative underperformance in certain scenarios can be attributed to a lack of specific mechanisms to handle domain shifts. (3) UDA methods (*e.g.*, DANN) exhibit superior performance in unsupervised domain adaptation tasks (*e.g.*, BM → B). However, their effectiveness is sometimes diminished in graph data, especially when dealing with complex datasets. (4) SLOGAN shows significant improvements in dataset splitting and cross-dataset scenarios, standing out, especially in cases where other methods falter. The improvement in accuracy reaches up to $2.3\%$ across these datasets.

### 4.3. Scalability Analysis

As shown in Figure 4, SLOGAN shows superior efficiency and performance. It shows SLOGAN's potential for practical applications where computational resources are limited.

### 4.4. Visualization

We use t-SNE to visualize causal and spurious features, as shown in Figure 5 and 6. We make the following observations: (1) Causal features are domain-agnostic. As shown in Figure 5, spurious features exhibit a significant bifurcation, aligning with either the source or the target domain. Con-

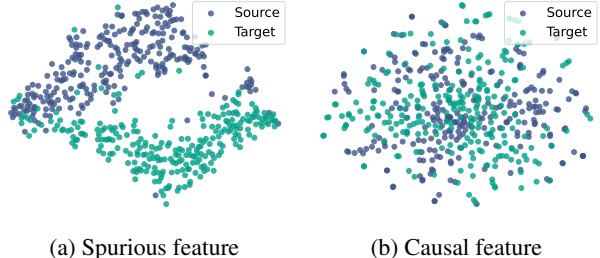

(a) Spurious feature   (b) Causal feature

*Figure 5.* Visualization of causal and spurious features distinguished by domains. Best viewed in color.

versely, causal features maintain a consistent distribution across both domains, underscoring their domain-agnostic nature. (2) Causal features are more aligned to semantic labels. As shown in Figure 6, the causal features are significantly partitioned according to positive and negative labels, illustrating their critical role in label prediction.

### 4.5. Ablation Study

Table 5 presents our ablation study results on the NCI1 dataset, detailing the model's performance without specific components. We find that each component: Disentanglement of causal and spurious features ($\mathcal{L}_{dis}$), unbiased discriminative learning using causal features ($\mathcal{L}_{sup}$), invariant learning under intervention ($\mathcal{L}_{inv}$). The absence of any one

*Table 5.* The Ablation Study classification results (in %) on NCI1 (source→target). N0, N1, N2, and N3 are sub-datasets.

| Methods | N0→N1 | N1→N0 | N0→N2 | N2→N0 | N0→N3 | N3→N0 | N1→N2 | N2→N1 | N1→N3 | N3→N1 | N2→N3 | N3→N2 | Avg. |
|---|---|---|---|---|---|---|---|---|---|---|---|---|---|
| Baseline | 55.8 | 59.1 | 54.0 | 73.3 | 65.0 | 70.7 | 73.5 | 60.7 | 70.2 | 67.8 | 54.5 | 55.1 | 63.3 |
| SLOGAN *w/o* $\mathcal{L}_{sup}$ | 71.0 | 59.7 | 53.9 | 72.3 | 72.3 | 69.4 | 73.8 | 69.4 | 71.4 | 67.8 | 58.7 | 69.9 | 67.5 |
| SLOGAN *w/o* $\mathcal{L}_{inv}$ | 70.4 | 61.2 | 63.1 | 71.4 | 72.3 | 70.9 | 75.3 | 71.8 | 72.3 | 74.9 | 59.8 | 70.4 | 69.5 |
| SLOGAN *w/o* $\mathcal{L}_{dis}$ | 71.1 | 63.7 | 61.3 | **73.5** | **73.8** | 71.4 | 75.3 | 72.3 | 71.4 | 71.6 | 59.3 | 70.0 | 69.6 |
| **Ours** | **71.4** | **64.1** | **63.1** | 71.8 | 72.3 | **72.8** | **76.7** | **72.5** | **73.3** | **76.3** | **61.7** | **70.9** | **70.6** |

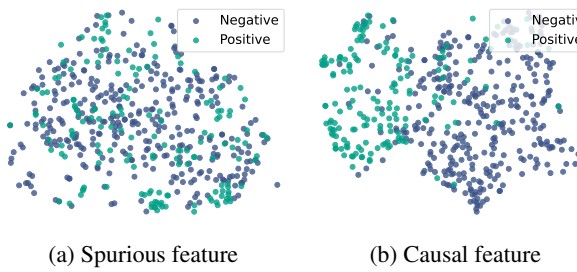

(a) Spurious feature      (b) Causal feature

*Figure 6.* Visualization of causal and spurious features distinguished by labels. Best viewed in color.

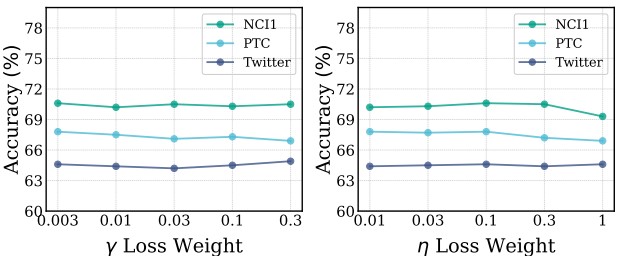

*Figure 7.* Sensitivity analysis on loss weight $\gamma$ and $\eta$.

component leads to a decrease in accuracy, with the removal of $\mathcal{L}_{dis}$ showing the most considerable impact, followed by $\mathcal{L}_{sup}$ and $\mathcal{L}_{inv}$. This shows the critical role each plays in the model's overall effectiveness.

### 4.6. Sensitivity Analysis

We test loss weight $\gamma$ and $\eta$, as shown in Figure 7. We varied $\gamma \in \{0.003, 0.01, 0.03, 0.1, 0.3\}$ and $\eta \in \{0.01, 0.03, 0.1, 0.3, 1\}$, indicate a marginal impact on the accuracy across different datasets. The results show that SLOGAN is robust to the hyper-parameters. The stability across a broad range of weights suggests that while the $\mathcal{L}_{sup}$ and $\mathcal{L}_{inv}$ components are essential, their performances are robust to weight variations. We set $\gamma = 0.003$ and $\eta = 0.1$ as the default value according to the experiments.

## 5. Related Works

***Graph classification*** (Ma et al., 2023; Jiang et al., 2023; Zhang et al., 2022) has emerged as fundamental task for structured data analysis with applications spanning social networks (Liu et al., 2021; Li et al., 2021), bioinformat-

ics (Borgwardt et al., 2005), and cheminformatic (Hansen et al., 2015; Kojima et al., 2020; Gilmer et al., 2017). Modern GNNs employ message-passing mechanisms with hierarchical pooling operations (Lee et al., 2019b; Wang et al., 2024b) for graph classification.

***Unsupervised domain adaptation*** (UDA) has become a pivotal approach for knowledge transfer from labeled source data to unlabeled target domains (Shi et al., 2022; He et al., 2022; Zhang et al., 2023; Pilancı & Vural, 2022), with extensive applications, *e.g.*, in computer vision (Long et al., 2018; Zou et al., 2018). The primary strategies in UDA include domain alignment (Yan et al., 2017; Lee et al., 2019a) and self-learning (Wei et al., 2021a; Xiao & Zhang, 2021). Self-learning methods, enhanced by semi-supervised learning techniques, aim to improve target domain performance through methods like pseudo-labeling (Sohn et al., 2020).

***Graph Domain Adaptation*** (GDA) (Ju et al., 2024) remains under-explored compared to its Euclidean counterparts. While node-level adaptation has seen initial progress (Wu et al., 2020; Zhang et al., 2021), graph-level adaptation faces unique challenges due to structural complexity and semantic richness (Zeng et al., 2024; Yin et al., 2023; Luo et al., 2024a). Existing approaches can be categorized into two main streams: (1) Global alignment methods (Dai et al., 2022; Shen et al., 2020) that directly minimize domain discrepancy through adversarial learning, but risk discarding crucial substructures while preserving spurious correlations; (2) Self-training approaches (Wei et al., 2021a; Luo et al., 2024b) that leverage pseudo-labels for target domain supervision, but suffer from error propagation and confirmation bias. These methods often struggle with feature entanglement between causal and spurious factors, leading to unstable adaptation performance. SLOGAN tackles these limitations through a novel perspective of sparse causal discovery and generative intervention.

***Causal Discovery & Disentanglement*** provides theoretical foundation for stable representation learning (Arjovsky et al., 2019; Zhao et al., 2023; Wang et al., 2024a) and stable learning (Yu et al., 2023). Recent works in graph learning (Chen et al., 2022; Yang et al., 2023; Cheng et al., 2024) combine causal inference with graph neural architectures, while disentanglement methods (Zhou et al., 2023) aim to separate domain-invariant factors. Our work ad-

vances this line by introducing sparse causal graphs and generative interventions specifically designed for graph-structured domain shifts. While IDEA (Wang et al., 2024a) applies causal disentanglement to image retrieval, SLOGAN uniquely addresses unsupervised graph domain adaptation through sparse causal modeling and a generative intervention based on cross-domain feature recombination.

# 6. Conclusion

This paper addresses causal-spurious feature entanglement and global alignment collapse in unsupervised graph domain adaptation. Our sparse causal modeling framework achieves stable knowledge transfer through innovations in causal graph construction, generative intervention mechanisms, and pseudo-label calibration. Extensive experiments demonstrate superior performance, with theoretical analysis proving our method's target error bound inversely correlates with causal feature retention. Future directions include theoretically quantifying intervention impacts, extending the framework to graph generation and temporal prediction, and exploring source-free graph domain adaptation, which has garnered increasing attention for its independence from source domain data.

# Impact Statement

This work advances unsupervised graph domain adaptation by addressing causal-spurious feature entanglement and alignment collapse through a novel sparse causal discovery framework. Our approach enables reliable knowledge transfer via causal graph construction, generative interventions, and pseudo-label calibration. We provide theoretical guarantees through target error bounds and demonstrate practical applications in molecular property prediction, social network analysis, and urban traffic modeling. By disentangling causal mechanisms from spurious correlations, our method enhances the reliability of graph learning systems in critical domains such as healthcare and autonomous systems.

# Acknowledgment

This paper is partially supported by grants from the National Key Research and Development Program of China with Grant No. 2023YFC3341203 and the National Natural Science Foundation of China (NSFC Grant Number 62276002).The authors are grateful to the anonymous reviewers for their efforts and insightful suggestions to improve this article.

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

# A. Algorithm

The overall algorithm of SLOGAN is summarized in Algorithm 1. We use the structural causal model to disentangle causal factors and spurious factors, with the information bottleneck principle. For causal factors, we employ an unbiased pseudo-labeling for discriminative learning on the target domain. For spurious factors, we utilize a generative model for invariant learning to mitigate their influence.

---

**Algorithm 1** Optimization Algorithm of SLOGAN

---

**Input:** Source dataset $\mathcal{D}^{so}$; target dataset $\mathcal{D}^{ta}$;
**Output:** GNN-based classifier $\Phi(\cdot)$;
 1: Pre-train $\Phi(\cdot)$ using $\mathcal{D}^{so}$;
 2: **for** epoch = 1, 2, $\cdots$ **do**
 3:     **for** each batch **do**
 4:         Sample mini-batch $\mathcal{B}^{so}$ and $\mathcal{B}^{ta}$ from $\mathcal{D}^{so}$ and $\mathcal{D}^{ta}$;
 5:         Generate confident target graphs $\mathcal{C}^{ta}$;
 6:         Generate reconstruction samples;
 7:         Calculate the loss objective using;
 8:         Update parameters of $\Phi(\cdot)$ by back-propagation;
 9:     **end for**
10: **end for**

---

# B. Proof of Target Domain Error Bound

**Theorem B.1.** *(Generalization Bound) Under a stable causal graph construction, assume the following conditions hold: (1)* **Causal Sufficiency**: *$I(Y; Z^c) > I_c$, where $Z^c$ is the causal variable and $I_c$ is an information contraint. (2)* **Spurious Suppression**: *$I(Y; Z^s) \leq \epsilon_1$, where $Z^s$ is the spurious variable. (3)* **Generative Intervention**: *$\mathbb{E}||Z - G(Z^c, Z^s)||_2^2 \leq \epsilon_2$, where $G$ is the generation model. Then, for any predictor $h \in \mathcal{H}$, with probability at least $1 - \delta$, the target domain error $\epsilon_T(h)$ is bounded as follows:*

$$\epsilon_T(h) \leq \hat{\epsilon}_S(h) + C\sqrt{\epsilon_1} + L\sqrt{\epsilon_2} + C(n_S, \delta), \tag{20}$$

*where $L$ is the lipschitz constant of the loss function, $C$ is a constant, and $n_S$ is the sample size in the source domain. Here, $\epsilon_T(h)$ represents the error in the target domain, while $\hat{\epsilon}_S(h)$ denotes the emprical error in the source domain.*

*Proof*: For any $h \in \mathcal{H}$, the source and target domain errors are defined based on $Z$, along with its causal and spurious components $Z^s$ and $Z^c$, which can be stated as follows,

$$\epsilon_S(h) = \mathbb{E}_{(Z,Y)\sim S}\left[l(h(Z), Y)\right], \quad \epsilon_S^{'}(h) = \mathbb{E}_{(Z,Y)\sim S}\left[l(h(G(Z^s, Z^c)), Y)\right], \tag{21}$$

$$\epsilon_T(h) = \mathbb{E}_{(Z,Y)\sim T}\left[l(h(Z), Y)\right], \quad \epsilon_T^{'}(h) = \mathbb{E}_{(Z,Y)\sim T}\left[l(h(G(Z^s, Z^c)), Y)\right]. \tag{22}$$

Since the loss function $l$ and the predictor $h$ are assumed to be lipschitz continuous, the gap between $\epsilon_S(h)$ and $\epsilon_S^{'}(h)$ is bounded as,

$$|\epsilon_S(h) - \epsilon_S^{'}(h)| \leq \mathbb{E}_{(Z,Y)\sim S}\left[|l(h(Z), Y) - l(h(G(Z^s, Z^c)), Y)|\right] \leq L\mathbb{E}_{(Z,Y)\sim S}||Z - G(Z^s, Z^c)|| \leq L\sqrt{\epsilon_2}, \tag{23}$$

where the second inequality follows from the lipschitz continuity property, and the third follows from Jensen's inequality. Similarly, we have $|\epsilon_T(h) - \epsilon_T^{'}(h)| \leq L\sqrt{\epsilon_2}$. Next, since $I(Y; Z^s) \leq \epsilon_1$, the total variation distance between $\mathbb{P}(Y|Z^s)$ and $\mathbb{P}(Y|Z)$ satisfies:

$$\mathbb{E}_{Z^s}\left[TV^2(\mathbb{P}(Y|Z^s), \mathbb{P}(Y))\right] \leq \frac{1}{2}\mathbb{E}_{Z^s}\left[D_{KL}(\mathbb{P}(Y|Z^s)|\mathbb{P}(Y))\right] = \frac{1}{2}I(Y; Z^s) \leq \frac{1}{2}\epsilon_1. \tag{24}$$

Here, the inequality follows from Pinsker's inequality, implying that the dependency of $Y$ on $Z^s$ is suppressed to $O(\sqrt{\epsilon_1})$. Under the causal sufficiency condition $I(Y; Z^c) > I_c$, the variable $Z^c$ is informative for predicting $Y$. Assuming the causal mechanism is stable across the source and target domains, the prediction based on $Z^c$ remains consistent. Hence, the primary

*Table 6.* Statistics of the datasets.

|  | Datasets | Graphs | Avg. Nodes | Avg. Edges |
|---|---|---|---|---|
| PTC | PTC_FM | 349 | 14.11 | 14.48 |
|  | PTC_FR | 351 | 14.56 | 15.00 |
|  | PTC_MM | 336 | 13.97 | 14.32 |
|  | PTC_MR | 344 | 14.29 | 14.69 |
| NCI1 |  | 4110 | 29.87 | 32.30 |
| TWITTER-Real-Graph-Partial |  | 144033 | 4.03 | 4.98 |
| Letter-Med |  | 2250 | 4.67 | 3.21 |

discrepancy between source and target domain errors arises due to $Z^s$, leading to, $|\epsilon'_S(h) - \epsilon'_T(h)| \leq C\sqrt{\epsilon_1}$. Combining this with inequality (equation 23), we obtain,

$$\epsilon_T(h) \leq \epsilon'_T(h) + L\sqrt{\epsilon_2} \leq \epsilon'_S(h) + C\sqrt{\epsilon_1} + L\sqrt{\epsilon_2} \leq \epsilon_S(h) + C\sqrt{\epsilon_1} + 2L\sqrt{\epsilon_2}. \tag{25}$$

Following the proof in (Rosenfeld & Garg, 2023), we can derive the statistical generalization bound, that is, with probability at least $1 - \delta$, the source domain error satisfies:

$$\epsilon_S(h) \leq \hat{\epsilon}_S(h) + \sqrt{\frac{log(1/\delta)}{2n_S}} \tag{26}$$

Thus, we conclude that the target domain error $\epsilon_T(h)$ is bounded as $\epsilon_T(h) \leq \hat{\epsilon}_S(h) + C\sqrt{\epsilon_1} + L\sqrt{\epsilon_2} + C(n_S, \delta)$. $\square$

## C. Extra Experimental Details

### C.1. Extra Dataset Details

Our study uses a diverse set of real-world datasets, encompassing areas from cheminformatics to social networks, to assess our unsupervised graph domain adaptation method. We employ both dataset division and cross-dataset strategies for validation. Below is information on these datasets, and statistics of the datasets are shown in table 6:

- *Cross-dataset scenarios*, which include PTC (Helma et al., 2001). The datasets are inherently unbiased across sub-datasets. We utilize their original splits as defined, ensuring a stringent test of our model's adaptability across different domains. This setup aims to validate our approach's effectiveness in handling variations inherent in separate datasets, each representing unique challenges regarding chemical structures or biological attributes.

- *Dataset-split scenarios*, which contain NCI1 (Wale & Karypis, 2006), TWITTER-Real-Graph-Partial (Pan et al., 2015) and Letter-Med (Riesen & Bunke, 2008) datasets. We follow previous works (Ding et al., 2018; Yin et al., 2022; Lu et al., 2023) to split the dataset by graph density, using their diverse nature to evaluate our domain adaptation capability. The datasets are divided into four subsets (*e.g.*, $T0, T1, T2$, and $T3$ for TWITTER-Real-Graph-Partial), organized by increasing levels of graph density. The split according to density introduces domain shifts, providing a framework for evaluating the effectiveness of our domain adaptation method.

Below is information on these datasets, and statistics of the datasets are shown in table 6:

- *Predicative Toxicology Challenge (PTC)* (Helma et al., 2001): Comprising chemicals for toxicology prediction, the PTC dataset contains compounds classified for carcinogenic potential in rodents. We categorize it into four groups: PTC_FM (female mice), PTC_FR (female rats), PTC_MM (male mice), and PTC_MR (male rats), each reflecting different domains based on the rodent's gender and species.

- **NCI1** (Wale & Karypis, 2006): Central to cheminformatics, the NCI1 dataset represents chemical compounds used in anti-cancer activity screening, especially for lung cancer. Graphs in NCI1 symbolize compounds, where nodes are atoms, edges are chemical bonds, and node attributes indicate atom types through one-hot encoding. Based on edge density variation, we divide NCI1 into four subsets, N0, N1, N2, and N3.

- **TWITTER-Real-Graph-Partial** (Pan et al., 2015): This dataset derives from Twitter for sentiment analysis, consisting of tweet-representative graphs. Here, nodes represent terms and emoticons, and edges signify their co-occurrence. We partition this dataset into T0, T1, T2, and T3, categorizing by edge density.

- **Letter-Med** (Riesen & Bunke, 2008): A collection of images comprising handwritten letters and medical documents. In these images, nodes represent the endpoints of strokes, while edges correspond to the lines connecting them. The letters exhibit significant distortion, which introduces complexity to recognition. This dataset is primarily utilized for optical character recognition and handwriting recognition tasks within the medical domain.

## C.2. Extra Baseline Details

In our study, we compare SLOGAN with an extensive selection of state-of-the-art baselines, spanning a wide array of strategies within the domain:

- **GIN** (Xu et al., 2018): GIN refines the Weisfeiler-Lehman graph isomorphism test to improve capture ability for complex graph topologies.

- **GCN** (Welling & Kipf, 2016): GCN leverages a simplified spectral graph convolution technique to merge node features.

- **GAT** (Veličković et al., 2018): GAT applies attention mechanisms to fostering a dynamic and locality-sensitive graph learning.

- **GraphSAGE** (Hamilton et al., 2017): GraphSAGE innovatively samples and aggregates neighborhood features, effectively adapting unseen nodes post-training.

- **Mean-Teacher** (Tarvainen & Valpola, 2017): Employing a student-teacher paradigm, this technique refines semi-supervised learning through prediction consistency, leveraging an ensemble of student models as the teacher.

- **InfoGraph** (Sun et al., 2020): Aiming to maximize mutual information across graph levels, InfoGraph generates potent graph representations under semi-supervised scheme.

- **DANN** (Ganin et al., 2016): DANN cultivates features that are relevant to the source task yet domain-agnostic, incorporating a gradient reversal layer to align domain distributions.

- **ToAlign** (Wei et al., 2021b): ToAlign applies task-oriented priors for a structured feature decomposition and alignment, bridging source and target domain disparities with finesse.

- **DUA** (Mirza et al., 2022): DUA leverages batch norm statistics to online test-time target data adaptation.

- **DARE-GRAM** (Nejjar et al., 2023): DARE-GRAM proposes to achieve cross-domain alignment with the inverse Gram matrix instead of the original feature.

- **CoCo** (Yin et al., 2023): CoCo contrast representation from different branch to improve the topology mining in the target domain, which is the latest state-of-the-art method in unsupervised graph domain adaptation.

