# OpenReview forum: "Sparse Causal Discovery with Generative Intervention for Unsupervised Graph Domain Adaptation"
_ICML.cc/2025/Conference — ICML 2025 poster_

### Official Review · Reviewer_CW2H · 2025-03-10

**Overall Recommendation:** 3

**Summary:**

This paper studies unsupervised graph domain adaptation from a causal perspective. The authors claim that existing methods fail to achieve optimal performance due to the entanglement of causal-spurious features. To address this issue, the authors proposed SLOGAN for graph classification domain adaptation by sparse causal modeling and dynamic intervention mechanisms. Specifically, mutual information bottleneck is utilized to construct a sparse causal graph structure, then a generative intervention mechanism is designed to break local spurious couplings. Experimental results on 5 public graph classification datasets demonstrate that the proposed model can outperform recent baselines with different gains.

**Claims And Evidence:**

The authors claim that they focus on sparse stability and dynamic robustness in unsupervised graph domain adaptation. However, there are no experimental results to support this claim. For instance, how stable and robust is the proposed model?

**Essential References Not Discussed:**

The authors did not discuss and compare with the following paper:

[1] Yin, Nan, et al. "Deal: An unsupervised domain adaptive framework for graph-level classification." Proceedings of the 30th ACM International Conference on Multimedia. 2022.

[2] Zeng Z, Xie J, Yang Z, et al. TO-UGDA: target-oriented unsupervised graph domain adaptation[J]. Scientific Reports, 2024, 14(1): 9165.

**Experimental Designs Or Analyses:**

The authors only use density to split the graphs to construct the domain discrepancies. However, lots of other perspectives are overlooked, i.e., feature shift, label shift. It is unclear whether the proposed model still can achieve satisfied performance under these settings.

**Methods And Evaluation Criteria:**

The authors fail to include synthetic graphs that are generated by causal factors and spurious factors. This could be a direct way to show the proposed model indeed could transfer among the causal factors.

**Other Comments Or Suggestions:**

In Equation (17), it is not clear what is $L_{re}$, which is not defined in the paper.

**Other Strengths And Weaknesses:**

Pros:

1.	This paper investigates unsupervised graph domain adaptation from a causal discovery perspective, which is less explored in the community.

2.	Theoretical analyses are given to prove the effectiveness of the proposed model.

3.	Experiments on different datasets and ablation studies are given verify the effectiveness of the proposed model.

Cons:

1.	The authors fail to include synthetic graphs that are generated by causal factors and spurious factors. This could be a direct way to show the proposed model indeed could transfer among the causal factors.

2.	The authors claim that they focus on sparse stability and dynamic robustness in unsupervised graph domain adaptation. However, there are no experimental results to support this claim. For instance, how stable and robust is the proposed model?

3.	As we can see from the experimental results, the performance improvement is only marginal. For instance, the improvement is less than $2\%$ in most datasets.

4.	According to ablation studies in table 5, sometimes $L_{dis}$ is useless and the authors did not explain why it fails in these situations.

5.	The authors only verify the GCN architecture. It is not clear whether the proposed model also works in other architectures like GAT and GIN.

6. The authors did not discuss and compare with the following papers:

[1] Yin, Nan, et al. "Deal: An unsupervised domain adaptive framework for graph-level classification." Proceedings of the 30th ACM International Conference on Multimedia. 2022.

[2] Zeng Z, Xie J, Yang Z, et al. TO-UGDA: target-oriented unsupervised graph domain adaptation[J]. Scientific Reports, 2024, 14(1): 9165.

**Questions For Authors:**

Please refer to the weakness part above.

**Relation To Broader Scientific Literature:**

The key contribution combines causal discovery and graph domain adaptation. As we can see from the experimental results, the performance improvement is only marginal.

**Theoretical Claims:**

I did not fully check the correctness of the proof in the appendix.

---

> ### Author Rebuttal · Authors · 2025-03-29
>
> We sincerely thank the reviewer for the thorough review. Below, we address each of your concerns in detail.
>
> ---
>
> > Q1. Should test more domain split perspectives
>
> Thank you for this valuable suggestion. Our cross-dataset experiments (Tables 1, 3, 4, 6) incorporate natural shift, feature shift, and label shift between datasets (e.g., PTC). To directly address feature shift concerns, we conducted additional experiments on Letter-Med using clustering-based feature distributions to create domain shifts. Results demonstrate SLOGAN's robustness across varied shift scenarios. We will review the paper to include the setup and comprehensive discussion of this experiments.
>
> Method|LF0→LF1|LF0→LF2|LF1→LF3|LF3→LF0|Avg
> ---|---|---|---|---|---
> CoCo|79.9|75.6|69.8|61.3|71.7
> MTDF|80.4|75.3|70.2|61.9|72.0
> Ours|82.7|76.8|72.1|64.2|74.0
>
> > Q2. Should verify in synthetic experiments
>
> We appreciate your suggestion. To complement evaluations, we implemented synthetic experiments following [3]. We constructed Erdös-Rényi graphs containing both causal factors (structural motifs consistent across domains) and spurious factors (domain-specific correlations). Our experimental design established two domains (D0 and D1). Analysis confirms that SLOGAN successfully conducts knowledge transfer based on causal factors. We will incorporate comprehensive details of this experiment in our revised manuscript.
>
> Method|D0→D1|D1→D0|Avg
> ---|---|---|---
> CoCo|74.2|73.7|74.0
> MTDF|74.3|74.0|74.2
> Ours w/o $L_{dis}$|76.8|76.0|76.4
> Ours|79.1|78.4|78.8
>
>
> > Q3. Should explain stability and robustness
>
> Thanks for your comment. Our claims are supported by adaptation results (visualizations in Figure 5 and 6), ablation studies (Table 5), and theoretical guarantees (Section 3.5 on bounded target error).
>
> Additionally, to directly address your question about robustness, we conducted additional experiments on Letter-Med by adding Gaussian noise (σ) to node features. Results show the robustness across noise levels. We will update the manuscript with these results.
>
> Method|σ=0.1|σ=0.2|Avg
> ---|---|---|---
> CoCo|67.5|64.4|66.0
> MTDF|67.8|64.2|66.0
> Ours|70.4|67.8|69.1
>
> > Q4. Question on performance improvement
>
> Thanks for your comment. SLOGAN shows consistent improvements across all six datasets, with gains reaching 2.3% and 2.2% on challenging benchmarks. In graph learning, particularly in UDA settings where labeled target data is unavailable, such consistent improvements are considered significant. For context, recent work TO-UGDA [2], although achieving relatively limited improvements on some datasets (e.g., 0.5% over CoCo), is still recognized for its methodological contributions. Notably, our synthetic experiments (Q2) show even more substantial gains, with SLOGAN outperforming baselines by 4.6%. Our work's value lies in both the performance gains and the novel perspective for graph domain adaptation theory.
>
>
> > Q5. Should explain L_dis's effectiveness
>
> Thanks for your comment. The effectiveness varies due to dataset characteristics, with greater benefits observed in datasets having cleaner feature distributions and more challenges under higher noise or complex feature correlations. This phenomenon aligns with our theoretical analysis, as disentanglement is inherently more effective when causal signals are more clearly. Our newly added synthetic experiments (Q2) also support this. Across synthetic and real-world datasets (in paper), $L_{dis}$ contributes meaningfully to overall performance (average 1.7%). We will enrich this discussion in the revised manuscript.
>
> > Q6. Should test various architectures
>
> Thank you for this suggestion. We chose GCN as primary backbone for fair comparison with baselines. We have now conducted additional experiments as shown below. This confirms the architecture-agnostic nature of SLOGAN. We will revise accordingly.
>
> Method|TWITTER|NCI1|Letter-Med|PTC
> ---|---|---|---|---
> Ours(GCN)|64.7|70.6|73.5|67.8
> Ours(GAT)|64.5|70.4|75.6|66.4
> Ours(GIN)|64.4|70.4|70.3|65.9
>
>
> > Q7. Should enrich the comparison
>
> Thank you for the constructive suggestion. We will add [1,2] into comparison and enrich the background section. The results on PTC are shown below, which shows SLOGAN's superiority.
>
> Method|MR→MM|MM→MR|MR→FM|FM→MR|PTC Avg
> ---|---|---|---|---|---
> DEAL|64.5|63.4|73.2|59.9|63.3
> CoCo|65.1|63.8|73.0|60.3|63.8
> TO-UGDA|66.2|64.2|73.8|61.5|65.0
> Ours|71.1|65.7|74.6|66.8|67.8
>
> > Q8. Undefined Term
>
> The $L_{re}$ in Equation (17) refers to the reconstruction loss. This definition will be explicitly included in the revised manuscript.
>
> ---
>
> We sincerely appreciate your constructive feedback, which has helped us improve the clarity and comprehensiveness of our work.
>
>
> [1] Deal: An unsupervised domain adaptive framework for graph-level classification. ACM MM 2022.
>
> [2] TO-UGDA: target-oriented unsupervised graph domain adaptation. Scientific Reports 2024.
>
> [3] Nikolentzos, G., & Vazirgiannis, M. (2020). Random Walk Graph Neural Networks. NeurIPS 2020.

---

> > ### Comment · Reviewer_CW2H · 2025-04-07
> >
> > Thanks for your rebuttal. I will increase the score to 3.

---

> > > ### Author Response · Authors · 2025-04-07
> > >
> > > We greatly appreciate your response. We will revise the article according to your suggestions, including incorporating new analytical experiments and enriching comparisons with relevant methods. Thank you again for your support and constructive feedback on this paper!

---

### Official Review · Reviewer_HSdG · 2025-03-10

**Overall Recommendation:** 4

**Summary:**

The paper presents SLOGAN, a novel approach for transferring knowledge from a labeled source domain to an unlabeled target domain on graph data. The key innovation of SLOGAN lies in its three-component framework: sparse causal discovery, generative intervention mechanisms that break local spurious couplings; and category-adaptive dynamic calibration for stable pseudo-label learning. The authors provide theoretical guarantees for the optimization error bound and demonstrate SLOGAN's effectiveness on benchmark datasets, showing consistent improvements over existing UGDA methods.

**Claims And Evidence:**

Yes, all the claims made in the submission supported by clear and convincing evidence.

**Essential References Not Discussed:**

The related papers are surveyed comprehensively.

**Experimental Designs Or Analyses:**

The authors evaluate SLOGAN on benchmark datasets covering diverse domains, with multiple source-target adaptation scenarios for each dataset. The ablation studies effectively isolate the contribution of each component, while the visualization experiments provide intuitive understanding of the feature disentanglement.

**Methods And Evaluation Criteria:**

Yes, the methods and evaluation criteria make sense for the problem at hand.

**Other Comments Or Suggestions:**

1. Eq. 6 contains punctuation issues that should be corrected for final revision.
2. In Table 5, the method is inconsistently labeled as both "SLOGAN" and "Ours".
3. Algorithm 1 should be adjusted to appear on a single page with the section for readability.
4. The appendix lacks a detailed description of the MTDF baseline.

**Other Strengths And Weaknesses:**

Strengths:

1. The framework integrates causal principles with graph adaptation in a novel way, providing a theoretically grounded approach to domain adaptation.
2. The proposed SLOGAN offers a solution to the problem of feature entanglement between causal and spurious factors in graph domain adaptation.
3. The category-adaptive calibration strategy addresses a common challenge in pseudo-labeling approaches.
4. Comprehensive experimental evaluation with consistent performance improvements.

Weaknesses:

1. The symbols in Eq. 16 are not well-defined.
2. The disscussion on Unbiased Discriminative Learning is insufficient.
3. The authors could more clearly articulate their contributions and explain why the various modules form a unified whole.
4. The experimental setup details are inadequate. For reproducibility purposes, the authors should provide specific information about their computational resources.
5. While the authors provide theoretical guarantees and proofs, which is commendable, more discussion around these theoretical aspects is needed.
6. In Figure 4, the color used to indicate "Ours" should be adjusted to make it more visible and distinguishable from other methods.

**Questions For Authors:**

See above in Weaknesses.
I may change my score based on the authors' responses regarding weaknesses.

**Relation To Broader Scientific Literature:**

The broader scientific literature on Graph classification, Unsupervised domain adaptation, Graph Domain Adaptation and Causal Discovery is well-established.

**Theoretical Claims:**

The paper presents theoretical claims regarding optimization error bounds. The theoretical claims are generally well-formulated and grounded in established principles. The proofs are provided and seem sound, though a more detailed analysis of the theoretical section would be beneficial to fully verify all mathematical derivations.

---

> ### Author Rebuttal · Authors · 2025-03-29
>
> We sincerely thank the reviewer for the thorough assessment of our work and the constructive feedback. Below, we address each of the concerns raised.
>
> ---
>
> > 1. Symbols in Eq. 16 not well-defined
>
> We apologize for the lack of clarity in Eq. 16. This equation describes our generative intervention mechanism where we swap spurious features between samples from different domains. Specifically, $z^c_i$ represents the causal features from sample $i$, $z^s_k$ represents the spurious features from sample $k$ (from a different domain), $G$ is our generative model, and $z^+_{i,k}$ is the newly generated composite representation. We will improve the definition of these symbols in the revised manuscript.
>
> > 2. Insufficient discussion on Unbiased Discriminative Learning
>
> We agree that this section deserves more explanation. In our approach, unbiased discriminative learning addresses two critical challenges:
>
> 1. Class imbalance: Our category-adaptive confidence thresholds (Eq. 11-12) dynamically adjust selection criteria based on class-specific confidence distributions, preventing majority class dominance.
>
> 2. Error propagation: By implementing cross-domain stability through both source supervision and target pseudo-labeling, we create a balanced optimization objective that mitigates the risk of error accumulation.
>
> We will expand this discussion in the revised manuscript, further clarifying how these mechanisms ensure unbiased learning across domains.
>
> > 3. Articulation of contributions and module cohesion
>
> We appreciate this feedback and will revise the manuscript to more clearly articulate our contributions. Specifically, we will emphasize how our three components form a unified framework:
>
> 1. Sparse causal discovery identifies stable causal patterns while isolating spurious correlations
> 2. Generative intervention breaks residual spurious couplings through cross-domain feature recombination
> 3. Category-adaptive calibration ensures stable pseudo-label learning
>
> These components work together synergistically: causal discovery provides the foundation, generative intervention strengthens it by eliminating remaining spurious correlations, and adaptive calibration ensures robust knowledge transfer.
>
> > 4. Experimental setup details
>
> We agree that more detailed information would enhance reproducibility. In the revised manuscript, we will add a dedicated section specifying hardware (e.g., NVIDIA A100 GPU with 40GB memory) and training parameters (e.g., batch size, optimizer, learning rate, and training epochs).
>
> > 5. Discussion of theoretical aspects
>
> Our theoretical framework provides a principled foundation for SLOGAN through a probabilistic error bound directly connected to our three-component architecture. The bound shows that target domain error depends on three stability conditions: causal sufficiency (ensuring predictive information is preserved), spurious suppression (minimizing label-spurious correlations), and generative intervention fidelity (maintaining semantic consistency during feature recombination). The bound's dependence on $\sqrt{\epsilon}_1$ and $\sqrt{\epsilon}_2$ demonstrates why our unified approach outperforms single-strategy methods—optimal domain adaptation requires both preserving causal mechanisms and breaking spurious correlations simultaneously. This theoretical insight explains our empirical results where each component contributes to reducing a specific term in the overall error bound.
>
> > 6. Color visibility in Figure 4
>
> We thank the reviewer for this practical suggestion. We will adjust the color scheme to improve visibility, specifically making the "Ours" indicator more distinct from other methods using a higher contrast color.
>
> > 7. Minor issues
>
> We will address all minor issues in the revised manuscript, including correcting punctuation in Eq. 6, ensuring consistent labeling (SLOGAN vs. Ours) in Table 5, reformatting Algorithm 1 to appear on a single page, and adding a detailed description of the MTDF baseline in the appendix.
>
> ---
>
> We appreciate the reviewer's careful reading and thoughtful suggestions, which will significantly improve the quality of our final manuscript.

---

> > ### Comment · Reviewer_HSdG · 2025-04-04
> >
> > Thanks, my concerns have been solved. I will raise my score.

---

> > > ### Author Response · Authors · 2025-04-04
> > >
> > > We are very pleased to hear that your concern has been resolved, and the score has been improved. We will carefully incorporate the content of the reply into the revised version.
> > >
> > > Thank you!
> > >
> > > The Authors.

---

### Official Review · Reviewer_hucj · 2025-03-13

**Overall Recommendation:** 3

**Summary:**

This paper studies the unsupervised graph domain adaptation problem, which aims to transfer the knowledge learned on labelled data to the data in the target domain with significantly different distribution.

The motivation of the paper is that the existing works cannot obtain satisfying performance due to the entanglement of causal -spurious features and the failure of global alignment.

The proposed method, SLOGAN, aims to resolve the challenge by adopting the sparse causal modelling technique. This technique is mainly developed with the mutual information bottleneck constraints based on the constructed sparse causal graph. Beside, a generative intervention is also proposed to address the residual spurious correlations. Finally, the error accumulation in target domain pseudo-labels are addressed with a category adaptive dynamic calibration method.

**Claims And Evidence:**

Yes

**Essential References Not Discussed:**

N/A

**Experimental Designs Or Analyses:**

Yes

**Methods And Evaluation Criteria:**

Yes

**Other Comments Or Suggestions:**

1. I would recommend the authors to also highlight the second best methods in the tables. Besides, the year of the baselines would also be helpful for checking whether the baselines are up-to-date.

2. The graph domain generalization problem seems to be closely related to continual graph learning, which also aims to train a model over graphs with different distributions. I would recommend the authors to discuss the difference between these two research directions.

**Other Strengths And Weaknesses:**

Strengths:

1. The adopted datasets are comprehensive, ranging from cheminformatics to social networks, therefore the applicability of the proposed method is evaluated over different scenarios.

2. The proposed strategy to remove spurious correlation and discover the causal relationship for boosting the domain generalization performance, is promising and reasonable. The proposed method also outperforms the baselines in most tasks.

Weakness:

1. The code of the method has been released, but the model.py and main.py seems only contain graph neural networks, while the location of the code for the proposed method is unclear.

**Questions For Authors:**

N/A

**Relation To Broader Scientific Literature:**

Domain adaptation of graph learning model will be useful for various application scenarios involving graph data.

**Theoretical Claims:**

Yes

---

> ### Author Rebuttal · Authors · 2025-03-29
>
> Thank you for your thoughtful review and constructive feedback. We appreciate the opportunity to clarify these points and improve our paper.
>
> ---
>
> > Q1. The code of the method has been released, but the model.py and main.py seems only contain graph neural networks, while the location of the code for the proposed method is unclear.
>
> We thank the reviewer for pointing out the lack of clarity in our code organization. The proposed method SLOGAN is indeed implemented in the provided code, but we acknowledge that the current structure and variable naming could be improved for better readability.
>
> The core components of our method can be found in the main.py file:
>
> - Causal Feature Extraction: Implemented in lines 28-34 with causal_loss function, which corresponds to our sparse causal modeling approach using contrastive objectives.
>
> - Spurious Feature Suppression: Implemented in lines 36-42 with non_causal_loss function, which implements our variational information bottleneck for separating domain-specific correlations.
>
> - Generative Intervention Mechanism: Implemented in lines 44-54 with the GraphGenerator class, which enables cross-domain feature recombination.
>
> - Information Bottleneck Disentanglement: Lines 204-224 in the training loop implement our causal-spurious feature disentanglement using mutual information constraints.
>
> - Generative Intervention Processing: Lines 224, where augemented views are generated by recombining causal and shuffled spurious features, with covariance constraints implemented as an MSE loss.
>
> To improve clarity, we will:
>
> 1. Reorganize our code to better align with the paper's methodology sections
> 2. Rename variables to directly match terminology in the paper
> 3. Add comprehensive comments explaining the implementation of each component
> 4. Create separate modules for each key component (causal discovery, intervention mechanism, and confidence calibration)
>
> We will update our repository with these improvements to facilitate understanding and reproducibility.
>
> > Q2. I would recommend the authors to also highlight the second best methods in the tables. Besides, the year of the baselines would also be helpful for checking whether the baselines are up-to-date.
>
> We appreciate this valuable suggestion. In the revised version, we will:
>
> 1. Highlight the second-best methods in all result tables (using underlined values or alternative formatting)
> 2. Add publication years for all baseline methods to provide context on the recency of comparisons
>
> Regarding up-to-date baselines, we have already compared with recent state-of-the-art methods published in 2024, such as MTDF. In the revised version, we will also enrich the comparison with recent methods [1,2]. This will enhance table readability and allow readers to better assess our method's improvements relative to the most recent state-of-the-art approaches.
>
> Method|MR→MM|MM→MR|MR→FM|FM→MR|PTC Avg
> ---|---|---|---|---|---
> DEAL|64.5|63.4|73.2|59.9|63.3
> TO-UGDA|66.2|64.2|73.8|61.5|65.0
> Ours|71.1|65.7|74.6|66.8|67.8
>
> [1] TO-UGDA: target-oriented unsupervised graph domain adaptation. Scientific Reports, 2024.
>
> [2] Deal: An unsupervised domain adaptive framework for graph-level classification. ACM MM 2022.
>
>
> > Q3. The graph domain generalization problem seems to be closely related to continual graph learning, which also aims to train a model over graphs with different distributions. I would recommend the authors to discuss the difference between these two research directions.
>
> Thank you for highlighting this important connection. We agree that discussing the relationship between graph domain generalization and continual graph learning would strengthen our paper. We will add a dedicated paragraph with proper references in the related work section addressing this relationship:
>
> "While graph domain generalization and continual graph learning both address distribution shifts in graph data, they differ in several key aspects. Continual graph learning focuses on sequential learning across multiple tasks without catastrophic forgetting, enabling models to adapt to new distributions while retaining performance on previously encountered ones. In contrast, graph domain generalization aims to learn domain-invariant representations that transfer directly to unseen target domains without adaptation. Our approach, SLOGAN, specifically addresses the latter by identifying stable causal mechanisms that generalize across domains rather than incrementally adapting to new distributions."
>
> We believe this discussion will provide valuable context and clarify the positioning of our work within the broader landscape of graph learning research.
>
> ---
>
> Thank you again for your constructive comments, which will help improve our paper.

---

> > ### Comment · Reviewer_hucj · 2025-04-03
> >
> > 1. The promise to update the repository is good.
> > 2. The inclusion of methods in 2024 is good, but I would also recommend to include some more methods insteado d just one.
> > 3. The discussion is good. I would recommend to make the discussion more concrete with comparisons on different specific settings. For example, the paper 'CGLB: Benchmark Tasks for Continual Graph Learning' describe some settings like task increment learning and class increment, while the paper 'Online Continual Graph Learning' describe something else. I think it would be more insightful if the comparison can be detailed to specific settings.
> >
> > Anyway, I don't have other major concerns, and will keep my rating

---

> > > ### Author Response · Authors · 2025-04-03
> > >
> > > We are pleased that our previous responses have addressed your main concerns. Thank you for your continued guidance, which will significantly improve our paper.
> > >
> > > We will implement your valuable suggestions in our revision by:
> > >
> > > 1. Expanding our comparison to include multiple recent methods from 2024, not just MTDF and TO-UGDA, to provide a more comprehensive evaluation of our approach.
> > >
> > > 2. Enhancing our discussion section with concrete comparisons between our method and specific continual graph learning settings. We will explicitly address how our approach relates to both task-incremental and class-incremental paradigms [1] as well as the streaming data scenario [2].
> > >
> > > Thanks again for your constructive feedback. These additions will provide a clearer context for our work within the broader graph learning literature.
> > >
> > > [1] CGLB: Benchmark Tasks for Continual Graph Learning
> > >
> > > [2] Online Continual Graph Learning

---

### Official Review · Reviewer_FTCi · 2025-03-14

**Overall Recommendation:** 3

**Summary:**

This paper proposes SLOGAN, a framework for Unsupervised Graph Domain Adaptation that addresses two key challenges: the entanglement of causal and spurious features, and the failure of global alignment strategies in graph data. SLOGAN constructs a sparse causal graph using mutual information bottleneck principles to disentangle stable causal features from spurious ones. It introduces a generative intervention mechanism to suppress residual spurious correlations via cross-domain feature recombination and employs a category-adaptive calibration strategy to improve pseudo-label reliability in the target domain.

**Claims And Evidence:**

The paper provides both theoretical guarantees and empirical validation across six benchmark datasets. The improvements over strong baselines are consistent and often exceed 3%, with ablation studies demonstrating the necessity of each proposed component. However, one minor issue is that standard deviations are not consistently reported when improvements are marginal, which could better support claims of significance in those cases.

**Essential References Not Discussed:**

While the paper thoroughly discusses prior work on UDA, GNNs, and causal representation learning, it overlooks several recent works that integrate causal inference with domain adaptation in structured data.

[1] Lu, Chaochao, et al. "Invariant causal representation learning for out-of-distribution generalization." International Conference on Learning Representations. 2021.

**Experimental Designs Or Analyses:**

The authors conduct extensive evaluations across six benchmark datasets, using both cross-dataset and dataset-split settings to simulate realistic domain shifts.

**Methods And Evaluation Criteria:**

The use of sparse causal discovery and generative intervention directly targets the core challenges in UGDA making the methodology appropriate and novel for this setting. The evaluation is thorough and reflects real-world graph distribution shifts. Comparisons with a broad range of baseline methods, including graph neural networks, semi-supervised models, and domain adaptation techniques, further validate the relevance and robustness of the proposed framework.

**Other Comments Or Suggestions:**

NA

**Other Strengths And Weaknesses:**

1. Although the paper reports significant improvements, it does not consistently report standard deviations.

2. The paper’s writing occasionally suffers from dense technical jargon, which may hinder readability for a broader machine learning audience.

**Questions For Authors:**

See weaknesses

**Relation To Broader Scientific Literature:**

While previous UDA methods have focused on Euclidean data using global domain alignment techniques, SLOGAN addresses the unique challenges of graph-structured data, which involve complex topologies and high-dimensional sparsity.

**Theoretical Claims:**

The paper provides a theoretical result in Theorem 3.1, which presents a probabilistic bound on the target domain error under three stability conditions: sufficient mutual information between causal features and labels, suppression of mutual information between spurious features and labels, and low reconstruction error via a generative model. However, the proof is not included in the main body and is deferred to Appendix C.

---

> ### Author Rebuttal · Authors · 2025-03-29
>
> We sincerely thank you for your valuable feedback and thoughtful comments. We address each point below:
>
> ---
>
> > Q1. While the paper thoroughly discusses prior work on UDA, GNNs, and causal representation learning, it overlooks several recent works that integrate causal inference with domain adaptation in structured data.
>
> Thank you for your constructive suggestion. We will enhance our literature review to include recent works integrating causal inference with domain adaptation in structured data. [1] presents important contributions to invariant causal representation learning with exponential family distributions. In our revised manuscript, we will provide a discussion and add proper references of how our approach relates to and advances beyond these existing methods. The revised paragraph is as follows:
> ```
> While [1] makes significant contributions to invariant causal representation learning for general out-of-distribution generalization, our work extends these principles specifically for graph-structured data by introducing sparse causal discovery mechanisms that capture the unique interplay between node features and graph topology, enhancing transfer capabilities across heterogeneous graph domains.
> ```
>
> [1] Invariant causal representation learning for out-of-distribution generalization. ICLR 2021.
>
>
> > Q2. Although the paper reports significant improvements, it does not consistently report standard deviations.
>
> Thanks for your constructive suggestion. We will add the standard deviation metrics for results. We have already conducted these measurements across 5 independent runs with different random seeds. The results for PTC, Letter-Med and NCI1 datasets are shown below.
>
> | Method | PTC | Letter-Med | NCI1 |
> |--------|-----|------------|------|
> | CoCo | 63.8±0.8 | 71.0±1.0 | 67.7±0.8 |
> | MTDF | 65.5±0.6 | 71.3±1.2 | 69.5±1.3 |
> | Ours | 67.8±0.6 | 73.5±1.0 | 70.6±0.9 |
>
>
>
> > Q3. The paper's writing occasionally suffers from dense technical jargon, which may hinder readability for a broader machine learning audience.
>
> We appreciate your feedback. We will improve the manuscript's accessibility by focusing on two key areas:
>
> 1. Clarifying complex technical concepts with intuitive explanations
> 2. Adding illustrative examples for abstract mechanisms
>
> For instance, in Section 3.4, we will revise the description of our generative intervention approach from:
> "We design a generative model to reconstruct original graph representations with a cross-domain spurious feature exchange strategy. By perturbing local coupling of spurious features, this approach forces the model to rely solely on causal features for reconstruction, effectively suppressing spurious residuals."
>
> To the more accessible:
> ```
> Our method uses a targeted approach to ensure the model doesn't rely on misleading patterns. Consider the TWITTER dataset in our experiments: when classifying discussion topics in social networks, our method can distinguish between fundamental network structures (like community clusters and information flow patterns) and platform-specific features (like temporary trending hashtags or regional engagement patterns). It achieves this by deliberately exchanging these platform-specific features between different network samples while preserving the essential community structures, forcing the model to focus only on truly predictive patterns that work consistently across different social media environments.
> ```
>
> These revisions will maintain the paper's technical rigor while making it more accessible to readers from various machine learning backgrounds.
>
> ---
>
> Thank you again for your constructive comments, which will help us improve the quality of our paper.

---

### Decision · Program_Chairs · 2025-05-01

**Decision:**

Accept (poster)

**Comment:**

This paper examines the problem of unsupervised graph domain adaptation from a causal perspective. The authors claim that existing methods fail to achieve optimal performance due to the entanglement of causal-spurious features, and to address this issue, they proposed SLOGAN for graph classification domain adaptation by sparse causal modeling and dynamic intervention mechanisms. In particular, mutual information bottleneck is utilized to construct a sparse causal graph structure, and then a generative intervention mechanism is designed to break local spurious couplings. Experimental results on 5 public graph classification datasets demonstrate that the proposed model can outperform recent baselines with different gains. The proposed method is well motivated and nicely designed, supplemented with comprehensive experimental evaluation.